# The Skin Microbiome: Current Landscape and Future Opportunities

**DOI:** 10.3390/ijms24043950

**Published:** 2023-02-16

**Authors:** Paisleigh Smythe, Holly N. Wilkinson

**Affiliations:** 1Centre for Biomedicine, Hull York Medical School, University of Hull, Hull HU6 7RX, UK; 2Skin Research Centre, Hull York Medical School, University of York, York YO10 5DD, UK

**Keywords:** microbiome, skin, ageing, senescence, wound healing, infection, antimicrobials, metagenomics, skin models

## Abstract

Our skin is the largest organ of the body, serving as an important barrier against the harsh extrinsic environment. Alongside preventing desiccation, chemical damage and hypothermia, this barrier protects the body from invading pathogens through a sophisticated innate immune response and co-adapted consortium of commensal microorganisms, collectively termed the microbiota. These microorganisms inhabit distinct biogeographical regions dictated by skin physiology. Thus, it follows that perturbations to normal skin homeostasis, as occurs with ageing, diabetes and skin disease, can cause microbial dysbiosis and increase infection risk. In this review, we discuss emerging concepts in skin microbiome research, highlighting pertinent links between skin ageing, the microbiome and cutaneous repair. Moreover, we address gaps in current knowledge and highlight key areas requiring further exploration. Future advances in this field could revolutionise the way we treat microbial dysbiosis associated with skin ageing and other pathologies.

## 1. Introduction

The skin is a highly dynamic organ composed of a range of cell types and structures that work together to preserve the cutaneous barrier and counter external challenges. The main layers are the epidermis and dermis, with underlying subcutaneous adipose tissue providing cushioning and energy reserves for the body [1]. The skin also houses appendages, such as hair follicles and glands, which are involved in many homeostatic functions, from thermoregulation to wound repair [2,3,4]. Sebaceous glands secrete non-polar lipids to prevent water loss [5], while antimicrobial peptides (AMPs) excreted from sweat glands limit the growth of pathogenic organisms [6,7]. Although the dermis preserves the structural integrity of the skin, the epidermis is the primary defence barrier maintaining direct contact with the extrinsic environment. This barrier comprises multiple biological, structural and chemical components crucial for preventing internal infection. However, perturbations to the cutaneous barrier, as occurs with skin ageing, pathology and injury, can cause microbial dysbiosis and increase infection risk [8,9,10]. Thus, this review will summarise existing knowledge of the dynamic interactions between the skin and its resident microbes, highlighting key disparities and future opportunities in this exciting field.

## 2. The Skin Barrier

The epidermis includes multiple heterogeneous layers, each performing a specialised role to preserve the skin barrier. The basal layer of the epidermis contains keratinocytes with stem-cell-like characteristics, which are attached to an underlying specialised matrix, the basement membrane [11]. A large proportion of the basal keratinocytes remain affixed to the basement membrane, but a subset of daughter cells progress through the epidermal layers via asymmetric mitosis [12]. This crucial mechanism enables self-renewal of the epidermis in a process known as terminal differentiation [13]. Daughter keratinocytes first enter the stratum spinosum, forming a layer of polyhedral-shaped cells joined together by desmosomes, intracellular junctions that mediate cell–cell adhesion and reinforce the epidermis against physical trauma [14]. Above the stratum spinosum is the stratum granulosum, a layer of flattened keratinocytes that form cytoplasmic keratohyalin granules to crosslink keratin filaments and create the water-impermeable barrier [15]. Finally, keratinocytes enter the external tier of the epidermis, the stratum corneum, where they release lysosomal enzymes that degrade their intracellular components [16]. This results in cells that are terminally differentiated, enucleated and tightly crosslinked to strengthen the skin barrier. As the stratum corneum is constantly shed and replaced every four weeks, the cycle of stratification is continuous and must be tightly controlled to prevent breaches to the skin surface [17,18].

Like other barriers of the body, the cutaneous barrier consists of microbial, immune, chemical and physical components [19]. However, unlike other epithelia, the skin exhibits an epidermal permeability barrier comprising the stratum corneum and a complex of tight junctions, adhesion proteins and cytoskeletal networks. Together, these structures prevent passive water loss from the body and protect against harmful chemical and biological agents [20]. This is apparent in investigations of epidermal function, where deficiencies in skin barrier proteins result in improper barrier formation, increased transepidermal water loss [21], reduced epidermal proliferation and differentiation [22,23] and skin barrier disorders [22,24]. Corneocytes of the stratum corneum are cemented together by a densely packed three-dimensional lipid matrix composed of ceramides, cholesterol and free fatty acids [5,25]. This matrix is formed from sebum produced by sebaceous glands, but also contains epidermal lipids and AMPs released from lamellar bodies of keratinocytes in the stratum granulosum [26]. The lipid layer of the stratum corneum safeguards the skin from desiccation by forming an impermeable barrier and provides a substrate for a range of microbial interactions (summarised in Figure 1). In particularly thick areas of the skin, such as the foot, there is an additional layer between the stratum granulosum and stratum corneum, the stratum lucidum, which acts as another impenetrable barrier to water [27].

The generation of free fatty acids on the surface of the skin creates a low-pH environment (pH 4–6) essential for barrier homeostasis [28]. An acidic skin pH is crucial for the activity of epidermal enzymes required for lipid processing and regulating cohesion proteins, thus preserving the stratum corneum and maintaining hydration levels [29,30]. Low skin pH also conserves the commensal skin microflora, which act as a first-line defence against pathogens through direct competition, influencing cutaneous immunity and supporting barrier homeostasis [31,32,33,34]. A range of immune cell subsets exist in the skin, secreting a plethora of cytokines and chemokines to modulate host response [35]. Indeed, skin colonisation with commensal bacteria shapes immunity through the activation of pattern recognition receptors, with the distinct activation signature dictating skin physiology. Activation of immune response pathways by commensal Staphylococcal spp. strengthens the immune barrier to prevent pathogenic infection [33], while commensals also trigger activation of Toll-like receptors (TLRs) to bolster immunity and accelerate wound repair [36]. Moreover, knockdown of certain pattern recognition receptors, such as NOD2, causes microbial dysbiosis and delayed wound healing [37], further demonstrating the importance of effective immune sensing in regulating cutaneous microbial colonisation. 

Keratinocytes likewise possess important innate immune functions, expressing TLRs and other immunomodulatory proteins (e.g., NOD2; [38]) that recognise pathogenic substances (pathogen-associated molecular patterns; [39]). Immune response pathways are activated in a TLR-specific manner, causing the release of cytokines, chemokines and AMPs to attract circulating immune cells. In addition to TLRs, keratinocytes constitutively express certain AMPs, such as HBD1, while others are induced in response to injury and infection [34,40,41]. Sebum lipids similarly contribute to immune surveillance, as many free fatty acids upregulate AMPs and cytokines in sebaceous glands [42,43,44] and display direct antibacterial properties [45]. Moreover, sebaceous-gland-rich sites of the skin harbour higher levels of AMPs (e.g., S100A7, DEFB4B and LCN2), chemokines and barrier genes than sebaceous-poor sites [46], suggesting that skin exhibits distinct immunological topography that could play a role in dictating the microbial composition of different skin sites.

TLRs are also required for barrier function as injury induces TLR3 to initiate inflammation [36], while activation of TLR2 elevates the expression of tight junction proteins (e.g., claudins and occludin) to enhance human skin barrier repair [47]. As TLRs are activated by microbial molecular signals, these findings indicate that resident microbes may be imperative for barrier maintenance and repair. This is certainly the case with other xenobiotic receptors; for example, mice deficient in the aryl hydrocarbon receptor exhibit an impaired skin barrier and are more susceptible to pathogenic colonisation with *S. aureus*. Further, colonisation of skin with a conglomerate of human skin commensals reduces transepidermal water loss and increases barrier gene expression in gnotobiotic mice [48]. Moreover, in humans, skin barrier perturbations (e.g., in atopic dermatitis) are associated with a lower abundance of protective skin commensals [8,49,50] and perturbations in TLR expression [47]. Despite these pertinent studies elucidating major roles of pattern recognition receptors in skin homeostasis, the vastly diverse landscape of host–microbe interactions remains largely unexplored due to the complexity of these interactions and the challenges associated with microbiome research (discussed in Section 8). 

**Figure 1 ijms-24-03950-f001:**
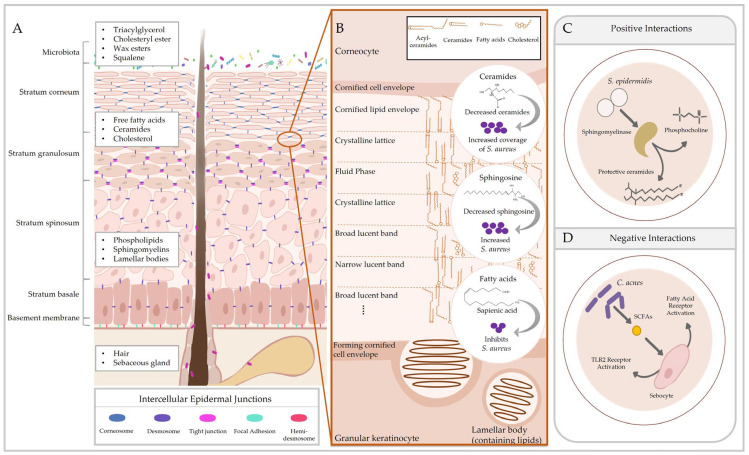
Cutaneous lipids preserve the epidermal permeability barrier and influence host–microbe interactions. The epidermis consists of multiple layers of phenotypically distinct keratinocytes that contribute to stratification. (**A**) Keratinocytes in each tier exhibit different adhesion structures responsible for maintaining skin integrity. Epidermal lipids are observed in the differentiating layers of the epidermis, while sebaceous lipids are secreted from sebaceous glands. (**B**) Molecular arrangement of the main stratum corneum lipids from [51] and effects of those lipids on *Staphylococcus aureus* colonisation [52,53,54]. (**C**) Skin microbiota also produce metabolites to utilise cutaneous lipids, generating products that contribute to barrier homeostasis (e.g., sphingomyelinase in *Staphylococcus epidermidis* [55]). (**D**) In pathological conditions, these interactions may negatively affect skin physiology (e.g., *Cutibacterium acnes* enhancing inflammation [56]). SCFA = short-chain fatty acids.

## 3. Skin Topography and the Microbiome

Maintenance of the cutaneous barrier is clearly critical to prevent pathogenic infection, with traditional skin microbiology focussing on prevention and treatment of infection by well-known pathobionts for this very reason [57]. Yet, the advent of gut microbiome research has now shifted our focus towards understanding the dynamic interactions that occur between the skin and its largely symbiotic community of bacterial, fungal, viral and Archaean inhabitants, collectively deemed the microbiota [58]. Our skin is second only to the gut in terms of bacterial density, with an approximate density of 10^4^ to 10^6^ bacteria per square centimetre and over 200 genera characterised [59]. The skin is home to 18 phyla, with 4 dominant ones: Actinobacteria (51.8%), Firmicutes (24.4%), Proteobacteria (16.5%) and Bacteroidetes (6.3%) [60]. Although skin microbiome research is still in its infancy, especially compared to that of the gut, pertinent studies have identified key roles for skin microbiota in maintaining homeostasis. These include providing nutrients (vitamin and amino acid synthesis [61,62]), inhibiting pathogenic growth [63,64], priming our immune system to differentiate between commensals and pathogens [65,66], and regulating epidermal differentiation [67]. A large proportion of the microbiome consists of resident microbes that are generally stable, but there is a smaller percentage of transient microbes that can opportunistically colonise niches when the skin is compromised [10]. Similar to the gut [68,69], the skin houses microbial communities that inhabit spatially distinct regions dictated by cutaneous topography (summarised in Figure 2; [70,71]), and while site-specific microbial composition is largely conserved, it can be affected by a range of other individual attributes, such as age, ethnicity, genetics, climate and skincare [71,72,73,74,75,76]. Skin diseases can also alter the microbiome and often present in a site-specific manner [10], indicating that exploration into the microbial habitation of ecological niches may provide significant insight into a range of skin pathologies.

The varied biogeography of the skin provides an unprecedented opportunity to explore how biological niches affect microbial composition. While the gut is rich in micro- and macronutrients that promote the growth of beneficial bacteria [77], the skin is a hostile environment with limited resources available. Thus, skin microbiota are specialised to utilise the chemical milieu of the stratum corneum, sweat and sebaceous glands. It is therefore not surprising that certain bacteria thrive in different skin regions, where there are varying levels of ultraviolet radiation (UVR) exposure, temperature, moisture, sebum content, oxygen availability and pH [78]. The skin microbiome shows the greatest conservation at higher taxonomic levels [56], and although bacterial communities group by skin type (sebaceous, dry and moist sites), fungi tend to segregate by body region alone [79]. 

The following seminal publications reveal the topographic and temporal distribution of skin microbiota, yet a major drawback of these studies is that they provide low taxonomic resolution, rarely classifying microbes beyond genus level. This is important given that the skin harbours a plethora of microbial species with broad phylogenetic representation [80,81]. Individual strains of bacteria also show differential associations with skin health and pathology, a prime example being species within the genus *Staphylococcus* [82,83,84]. Consequently, future studies should shift their focus to understanding strain-level temporospatial diversity of the skin.

Sebaceous (oily) sites (e.g., the torso, back and face) are highly acidic due to the abundance of free fatty acids [28]. These regions are predominantly inhabited by bacteria that can metabolically utilise sebum and tolerate low pH, such as *Cutibacterium* (formerly *Propionibacterium* [60]). As *Cutibacterium* spp. require anaerobic growth conditions, they are found in the pilosebaceous units of sebum-rich areas, where they produce lipases to convert sebum triglycerides into short-chain fatty acids, including propionic acid [85]. This is further supported by the fact that facial sebum levels directly correlate with *Cutibacterium* prevalence [86]. *Staphylococcus* is the second most predominant genus in the sebaceous skin microbiome. Staphyloccocci, such as *Staphylococcus aureus* and *Staphylococcus epidermis*, are tolerant of the acidic pH found in oily skin and produce lipases to utilise the lipid-rich substrate of these sites [32,87,88]. Interestingly, the retroauricular crease is a sebaceous site with low phylotype richness that tends to remain temporally stable due to the prevalence of *Cutibacterium* [78,89].

Moist sites (e.g., antecubital fossa, inguinal crease and popliteal fossa) are areas with a higher temperature and humidity, and a variety of hair follicles and glands [90]. The moist niche provides a myriad of nutrients, such as salts, sterols, esters and lipids, enabling the growth of *Staphylococcus* and *Corynebacterium* [60]. Corynebacteria are dominant colonisers of warm and moist environments [91], while the halotolerant *Staphylococci* are found at high density in moist, salt-rich sites such as sweat glands [92]. Although the microbiota of moist skin remains relatively stable, shifts in diversity can be observed between individuals [93,94]. By contrast, dry sites (e.g., hypothenar palm and volar forearm) exhibit high microbial diversity and low temporal stability [60,70,94]. Interestingly, different studies display variability in the dominant bacteria reported in dry skin, which may be influenced by bacterial biomass and temporal stability. For example, Flavobacteriales and β-Proteobacteria were dominant in Grice et al. [60], but *Cutibacterium* showed the highest contribution to dry sites in Oh et al. [70].

**Figure 2 ijms-24-03950-f002:**
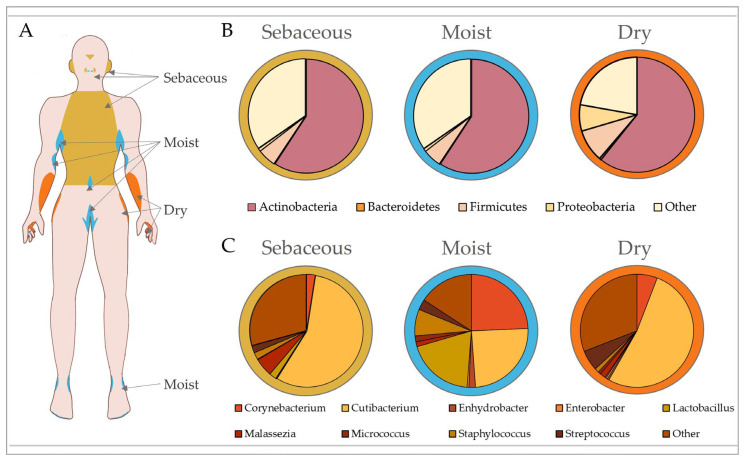
Microbial composition of the skin is dictated by topography. Skin exhibits biogeographically distinct regions that are generally categorised according to their unique physiological characteristics into sebaceous, moist and dry sites (**A**). These regions exhibit different microbial community structures influenced by skin pH, temperature, sebum content and moisture levels. Microbiota are separated by phylum according to site-specific variation in (**B**), with top contributing genera shown in (**C**). In healthy adults, sebaceous (e.g., torso, glabella and retroauricular crease) and dry (e.g., volar forearm and hypothenar palm) sites are predominantly colonised by *Cutibacterium*, while moist sites (e.g., antecubital fossa, inguinal crease and axillary crease) show equally high abundances of *Corynebacterium*, *Cutibacterium* and *Staphylococcus*. Data analysed from Oh et al. [94].

Although the microbiota consists of a wide range of microorganisms, most microbiome research has focussed on characterising the topography and function of bacteria, with less consideration given to our fungal, viral and Archean inhabitants due to their generally lower abundance [79,95]. Nevertheless, there have been some advancements in profiling the skin’s fungal microbiome (or mycobiome). As with bacteria, fungi reside within each skin niche, with community structure linked to skin region and age-related changes (reviewed in [96]). Culture-based methods initially identified *Malassezia*, *Rhodotorula*, *Debaromyces*, *Cryptococcus* and *Candida* as commensal skin flora [97,98,99]. These findings have since been extended using amplicon sequencing to reveal site-specific variation in fungal communities, where there is a general predominance of *Malassezia*. Additionally, the foot mycobiome is vastly more diverse than that of other body sites and includes *Malassezia*, *Aspergillus*, *Cryptococcus*, *Rhodotorula* and *Epicoccum* [79]. Further investigations are required to fully characterise fungal residency on the skin and provide insight into the interactions between fungal, bacterial and host communities.

## 4. The Skin Microbiome throughout the Life Course

Understanding normal skin physiology and microbial interactions enables greater appreciation of homeostatic mechanisms but also provides insight into how perturbations to normal cutaneous function are associated with disease. Indeed, skin topography, function and microbial composition are not only dictated by body site but can be influenced by a range of endogenous (e.g., genetics, age and gender) and environmental (e.g., lifestyle, climate, air pollution and use of cosmetics) factors [78,100,101]. The inherent high complexity and variability of the skin microbiome makes it challenging to delineate its role in specific physiological processes [102]. Despite this, several pertinent studies have shed light on how the skin microbiota change throughout the life course [103,104,105], and how perturbations in microbial composition are associated with skin disease and ageing [106,107,108].

Even though first microbial contact is thought to begin in utero [109], microbial colonisation rapidly expands at birth with exposure to the atmospheric environment [110]. Initial microbial colonisation is limited to a particular set of bacterial taxa from the birth canal, but is dictated by delivery route, as babies born via caesarean section exhibit microbiome profiles that more closely mimic the mother’s skin (dominated by *Staphylococcus*, *Corynebacterium* and *Cutibacterium*) than vagina (dominated by *Lactobacillus* and *Prevotella* [111]). Birth mode also affects the relative abundance of specific fungal spp., with vaginal births resulting in higher levels of *Candida albicans* [112]. However, by six weeks of age, microbial composition cannot be distinguished by delivery route [113]. Additionally, the vaginal microbiome changes throughout gestation, with pregnant mothers displaying lower bacterial diversity and higher contribution of *Lactobacillus* compared to non-pregnant women [114,115] and women postpartum [116]. 

Initial skin microbial colonisation is responsible for shaping the successional transition of the microbial ecosystem into adulthood. This first contact, alongside pre-programmed mechanisms, is crucial for the development of infant immunity [117], which preferentially develops tolerance to skin commensals [66,118]. It has been postulated that disruption of native microbial transfer, as observed with C-section, heightens the risk of type 1 diabetes due to links between the microbiota and immune system [119]. However, metanalyses assessing these risks have found mixed results [120,121]. By contrast, disrupting microbiota during pregnancy (e.g., with antibiotic use) may contribute to disease risk in offspring. This is certainly true in vivo, as vancomycin given to pregnant does markedly shifts gut microbial composition and increases asthma severity in murine offspring [122,123]. Thus, an appreciation of microbiome development in early life provides insight into the associations between microbial dysbiosis and pathology, proposing that the skin microbiota could be harnessed to improve health.

The microbiota of new-borns is undifferentiated across different body regions [111,113], but site-specific changes begin to occur within three months of age, with microbial diversity stabilising over the first year of life [113,124]. The diversity of the skin microbiome continues to adapt throughout childhood [125] and adolescence [126] and is shaped by the changing microenvironment of different skin sites during sexual maturation [127]. For example, both *Cutibacterium acnes* and *S. epidermidis* increase in a site-specific manner upon sexual maturity in males and females, while the lipophilic *C. acnes* and *Malassezia restricta* positively correlate with higher concentrations of certain sex hormones in female children (estrone, 17β-estradiol and testosterone [127]). These alterations are likely influenced by the effects of hormones on the skin, such as androgens increasing sebaceous gland activity [128]. Moreover, certain *C. acnes* strains are predicted to produce excessive porphyrin, which can cause skin inflammation and contribute to development of acne vulgaris [129]. These findings may perhaps shed light on why certain microbiome-linked skin pathologies are dependent on life stage.

In adults, the microbiota of the skin remains relatively stable, but as discussed previously, is highly dictated by the skin’s topography [94]. It has even been suggested that the observed stability of the adult skin microbiome could be used to predict an individual’s chronological age within four years [130]. Adult skin displays higher levels of *Cutibacterium* and *Corynebacterium*, while young children show a dominance of Gammaproteobacteria and Streptococcaceae at multiple sites [131]. These shifts in microbial composition coincide with physical changes in the skin, as the skin of infants is thinner, more alkaline and has a higher cell turnover rate compared to adults [132,133,134]. It is therefore not surprising that naturally aged skin, featuring many physiological and structural modifications, exhibits drastic compositional alterations in the skin microbiome [76,104,105,134]. 

## 5. Skin Ageing and the Microbiome

The skin is unique compared to other organs in that skin ageing is dictated by intrinsic and extrinsic factors. Sun-protected sites of the body, such as the buttocks, largely undergo intrinsic ageing processes influenced by genetic, metabolic and hormonal changes (e.g., reduction in 17 β-estradiol; [135]). Intrinsically aged skin is characterised by reduced sebaceous gland function, decreased blood flow and degradation of collagenous and fibrous extracellular matrices (ECMs), leading to atrophy, reduced lipid content, xerosis and fine lines [136,137]. By contrast, extrinsic ageing is triggered by environmental factors but is predominantly dictated by exposure to UVR [138]. Therefore, extrinsically aged skin is commonly found in sun-exposed sites of the body, such as the face and hands, and is depicted by telangiectasia, hyperpigmentation, deep wrinkles and a leathery appearance [139]. Both intrinsically and extrinsically aged skin have a higher pH, less hydration and reduced expression of tight junction proteins compared to young adult skin, but unlike intrinsic ageing, photoageing causes elevated proliferation and increased sebum [136,140]. Certain skin constituents that are altered with age (e.g., lipids) are also linked to microbial surveillance and skin barrier homeostasis [42,141]; hence, it is unsurprising that aged skin features marked dysbiosis in microbial composition (summarised in Figure 3). 

Intrinsic and extrinsic ageing trigger differential structural and functional alterations to the skin, yet both processes cause similar age-related changes in species richness compared to young adult skin [106]. As intrinsic ageing underpins both types of ageing, these results suggest that microbial diversity is highly dictated by intrinsic ageing mechanisms. This has been corroborated in other studies in women, with the within-sample microbial diversity of aged skin being significantly higher than that in young adult skin and most significant in sun-protected sites [9,104,108,141]. Overall abundance of bacteria increases with age, but this is not directly proportional as certain bacteria become more dominant (e.g., *Corynebacterium*) while others decline in number (e.g., *Cutibacterium* and *Lactobacillus* [9,104,108,141]) in a site-independent manner. Certain genera are also altered by body site, such as higher abundance of *Streptococcus* and lower abundance of *Staphylococcus* on the buttocks [9]. Age-related mycobiome shifts include reduced *M. restricta* and greater abundance of other fungal spp. on cheeks [104].

Several studies have performed correlative analyses to attempt to link the age-associated alterations in the skin microbiome to the structural and physiological changes that occur with intrinsic and extrinsic skin ageing [9,86,104,141]. Indeed, Howard et al. [9] demonstrated that ageing decreases the facial sebaceous gland area and increases the number of ceramides, lipids and natural moisturising factors, which positively and negatively correlate with specific bacterial genera. Moreover, Kim et al. [104] revealed that the dominant metabolic pathways of aged skin bacteria were linked to fatty acid degradation, antibiotic biosynthesis and bacterial motility. However, it is difficult to directly link the observed correlations between microbial dysbiosis and skin ageing without undertaking mechanistic investigations. Many of these studies are simultaneously constrained by utilising small cohorts, one gender and one ethnic group, making it difficult to generalise findings. Moreover, these studies often do not identify microbes beyond genus level, yet it is known that individual strains of bacteria are linked to age-associated skin pathology [82,83,84,142]. Future investigations should therefore aim to characterise age-related changes in the microbiome using more sophisticated techniques, appropriate study cohorts and selecting skin sites that enable delineations between intrinsic and extrinsic ageing.

A handful of pertinent investigations have begun to elucidate how age-related alterations in bacteria can affect host fitness; it has been suggested that age-specific dominant microbes can influence immunity and inflammation [108,143] and may regulate the intrinsic ageing process (reviewed in [144]). Notably, preventing the age-related decline in gut compartmentalisation limits microbial dysbiosis and extends lifespan [145]. In the skin, age-related changes in microbial community structure could drive pathogenic colonisation, contributing to many of the deleterious effects described above. In fact, *Cutibacterium*, which is reduced in aged skin, produces free fatty acids and thiopeptide antibiotics to suppress the growth of methicillin-resistant *S. aureus* and group A *Streptococcus* [146,147]. Other commensal bacteria, such as *Bacillus* and coagulase negative *Staphylococcus*, can also produce antimicrobials or induce the skin’s innate immune response to prevent pathogenic growth [33,34,148]. Nevertheless, our knowledge of age-related shifts in microbial taxonomy remains enigmatic, making it difficult to extrapolate the associated functional consequences for the skin.

## 6. Cellular Ageing and Microbial Dysbiosis

Cellular senescence (cellular ageing) is an area of research gaining considerable traction for its role in intrinsic skin ageing and potential for therapeutic targeting [149]. Cellular senescence is a mechanism whereby cells undergo transient or permanent cessation of proliferative capacity in response to intrinsic and/or extrinsic stressors linked to the ageing process, such as DNA damage, oxidative stress, mitochondrial dysfunction, inflammation and telomere shortening [150]. Senescent cells remain viable yet have significant metabolic and genetic alterations compared to their non-senescent counterparts, including the reorganisation of chromatin [151], increased lysosomal activity [152,153] and activation of a DNA damage response [154]. The central dogma is that senescence evolved to suppress tumours in young organisms by preventing neoplastic growth [155]. However, senescent cells accumulate with increasing physiological age, contributing to a range of age-related diseases by virtue of their functional perturbations and unique secretome (or senescence-associated secretory phenotype, SASP [156]. The SASP includes a range of secreted cellular products (e.g., growth factors, cytokines, chemokines, proteases and lipids [156,157,158]) that alter the tissue microenvironment in a context-dependent manner. For example, in wounds, a transient SASP mediated by PDGF-AA enables effective deposition of ECMs during skin repair [159], while a chronic SASP governed by CXCR2 delays healing in diabetes [160]. SASP genes are known to be upregulated at the transcriptional level by NF-κB and C/EBP β, but other pathways are also involved (e.g., mTOR) [161,162]. Moreover, the SASP can induce senescence in a paracrine manner, thus exacerbating tissue destruction and ageing phenotypes [163,164,165].

Aged skin is characterised by high levels of senescence in the epidermis, dermis, skin appendages and subcutaneous tissue [166,167,168,169]. Dermal fibroblasts from intrinsically aged skin also produce a secretome including canonical SASP factors (e.g., IL-1B, MIF and PAI-2 [156]), and proteins that may be unique to skin ageing [170]. Senescent cell accumulation in aged tissues is partially driven by age-related decline in the innate and adaptive immune systems, which are required to effectively clear senescent cells [171]. In addition, senescent cells are able to evade immune clearance by expressing antigens to inhibit natural killer cell activation and avoid T-cell recognition [172,173]. More recently, it has even been suggested that senescent fibroblasts can evade immune detection by secreting specific lysophospholipids [158]. Tissues with high levels of senescence display reduced functionality, increased inflammation and degradation of the ECM, due in part to the excessive production of matrix metalloproteinases (MMPSs) [174,175,176]. The fragmentation of ECM components, such as collagen and elastin, is clearly apparent in aged dermis exhibiting high levels of senescence and MMPs [177,178,179]. Multiple authors have demonstrated that catalase activity is reduced in intrinsic and photoaged skin, leading to elevated oxidative stress, MMP1 expression and collagen fragmentation [180,181]. This fragmentation of collagen even stimulates MMP1 production in fibroblasts cultured in vitro [181] to cause further ECM breakdown.

Though few direct links have been made between senescent cell accumulation and the physical characteristics of skin ageing, it is established that senescent fibroblasts cause hallmark signs of skin ageing, such as reduced epidermal thickness and impaired barrier formation in vitro [182] and in 3D skin equivalent models [183,184]. We know far less about the role of epidermal senescence in skin ageing, with most studies in this area focussing on the effects of UV exposure on senescence induction in keratinocytes and fibroblasts [185,186,187]. Interestingly, epidermal senescence elevates skin barrier permeability [188], which could be instigated by disruption of tight junctions [189,190], loss of structural proteins [191] and aquaporins [192] and decreased production of the enzymes responsible for sphingolipid synthesis [193]. In the epidermis, melanocytes are the primary cell type expressing the canonical senescence marker, p16ink4a, during skin ageing [166,194], and they contribute to skin atrophy in 3D skin equivalents via a CXCR3-governed SASP [164]. Indeed, these studies reveal that senescence drives skin barrier defects, which may impact microbial dysbiosis during skin ageing.

While microbiome changes in the skin correlate with age-associated alterations in cutaneous physiology [9,104], functional links between the skin microbiota and cellular ageing remain to be investigated. Most pertinent studies linking the microbiota to cellular senescence involve the gut, demonstrating that microbial dysbiosis enhances senescence and contributes to age-related disease states [195,196]. Moreover, ablation of senescent cells using senescence-targeted drugs (senolytics) alters gut microbial composition and ameliorates ageing pathology in mice [197,198], whereas a healthy gut microbiome is directly linked to healthy ageing in humans [199]. Notably, commensal *Cutibacterium* spp., which are depleted in aged skin, produce antioxidants capable of protecting the skin against cellular ageing [200], while genotoxins produced by pathogenic bacteria readily induce senescence in T cells [201]. Collectively, these findings suggest an important therapeutic link between the microbiome, cellular senescence and age-related pathology that could extend to the skin. Future investigations should therefore begin to explore this area, perhaps by identifying the microbial drivers of age-related changes in the skin using longitudinal metagenomic characterisation, followed by mechanistic assessment of the role of these “age-related” bacteria on the skin barrier and dermal matrix. Moreover, studies should be undertaken to explore how manipulation of senescence (e.g., by using murine ageing models and senolytics) may alter the skin microbiome and affect skin function. This will also have important implications for skin integrity and wound repair, which is discussed below. 

## 7. Wound Pathology, Ageing and Infection 

Breaches of the skin barrier expose subcutaneous tissue to microbial colonisation. In young, healthy individuals, closure of this barrier occurs rapidly via an orchestrated series of cellular events [202]. However, in ageing and related pathologies, skin repair is severely delayed [203], increasing risk of infection and the development of chronic, non-healing wounds [204,205,206]. Chronic wounds are a major socioeconomic burden, costing UK and US healthcare providers billions to treat annually [207,208]; thus, there is an urgent unmet need to fully elucidate the factors contributing to age-related delayed skin healing to develop new and effective therapies. In normal healing, a short-term, transient senescence is required to enable effective deposition of new connective tissue [159] and prevent excessive fibrosis [209]. However, in poor-healing wounds, there are high levels of infection and inflammation that prolong senescence and promote tissue breakdown [160,210,211,212]. Further, we have demonstrated that senescence in diabetic wounds is directly linked to healing outcome, as the ablation of senescence significantly accelerates wound repair in mice [160], while others show that levels of senescence may predict healing in human chronic wounds [212].

Aged skin is more susceptible to infection, in part because the immune system becomes senescent [213]. Altered microbial composition could also increase infection risk following injury, due to the higher numbers of pathobionts [82,214] and lower levels of commensals known to protect against pathogens by modulating host immunity and inhibiting bacterial virulence mechanisms [33,146,215]. This extends to elderly patients with diabetes who are at high risk of developing diabetic foot ulcers. These patients exhibit reduced expression of AMPs, such as RNAse 7 [216], and an abundance of virulent strains of *S. aureus* on their feet, both of which can independently increase infection risk [217]. Interestingly, oxidative stress (a key inducer of senescence) may additionally promote microbial dysbiosis in ageing and diabetes [218].

Recalcitrant wound infection is a key contributor to poor healing in the elderly that is exacerbated by high levels of antimicrobial resistance (AMR) and the presence of bacterial biofilms [219,220,221,222]. Vascular insufficiency is another important factor linked to infection in chronic wounds, decreasing the influx of immune cells and reducing efficacy of systemic antibiotics [223]. In addition, vasculopathy and neuropathy mask hallmark signs of infection, such as pain and erythema, making it difficult to diagnose infection at an early intervention stage [224]. Despite these risks, no studies to date have correlated changes in the skin microbiome with vasculopathy or neuropathy. 

In normal healing, interactions between commensal bacteria (e.g., *S. epidermidis*) and host immunity are necessary to mediate effective repair [225], yet impaired host response leads to microbial dysbiosis and delayed healing in mice [226]. Many reports have measured the microbial composition of chronic wounds [227,228] and correlated microbial composition to healing outcome [82,229,230,231,232]. However, these investigations rarely mechanistically demonstrate the functional links between microbial dysbiosis and healing. Due to the compromised skin barrier, wounds show a higher contribution of opportunistic pathogens, such as *S. aureus*, and lower amounts of commensal *Staphylococcus* and *Corynebacterium*, than intact skin [82,231,232]. Non-healing chronic wounds also exhibit overrepresentation of facultative anaerobes, such as *Enterobacter* and *Proteus*, compared to wounds that heal within six months [232]. Moreover, the recent observations that specific strains of *S. aureus* are associated with poor healing outcomes [82] and infection [83], whereas others bolster host response and skin regeneration [233,234], highlight the need for subspecies-level profiling to elucidate the microbial drivers of poor healing in the elderly.

## 8. Challenges in Skin Microbiome Research 

### 8.1. Microbial Identification

Traditional microbial research focusses on understanding the interactions between the host and individual pathogenic organisms in isolation. However, the emerging evidence that the microbiome plays important roles in health and disease has created a need to (1) accurately identify the microbes that inhabit biological niches and (2) enable collective profiling of native microbial communities [235]. Nevertheless, developing identification techniques that can suitably cover the breadth and depth of our microbial ecosystems is a major challenge in microbiome research. Traditionally, microbial communities have been explored primarily using culture-based methods, which select for species capable of growing in artificial laboratory conditions and greatly underrepresent the diversity of the native microbial environment [236,237]. Indeed, culture-based assays can underestimate the bioburden of diabetic foot ulcers by up to 26 bacterial species and fail to identify *S. aureus* in over 50% of samples [238]. Various 16S techniques have also been found to be advantageous over culture-based protocols for analysing wound swabs and biopsies, identifying around 50% more bacterial species, including obligate and facultative anaerobes [238]. Despite the poor resolution provided by culture-based practices, they are still necessary to enable full exploration of host–bacteria interactions, and their combination with system-level approaches, such as omics technologies [239] and computer modelling [240,241], can provide crucial insight into microbial diversity and function. 

Over thirty years ago, Woese and Fox [242] pioneered community non-culture-based profiling to circumvent some of the challenges associated with traditional culture techniques. They demonstrated that targeted amplification of highly conserved 16S ribosomal RNA (rRNA) genes could be utilised for taxonomic identification because they include hypervariable regions that can be used to classify bacteria based on sequence homology [243]. The 16S rRNA sequencing approach remains the most widely utilised method in microbiome research to date, but is not without its disadvantages. Biases can be introduced by the PCR amplification methods and sequencing platforms used, resulting in inaccurate estimates of microbial diversity [244,245,246]. Moreover, 16S rRNA sequencing provides limited resolution, being rarely capable of classifying bacteria beyond the genus level. This is an important consideration for skin research, as bacteria within the same genus can stimulate differential effects on the host. For instance, atopic dermatitis patients with a predominance of *S. aureus* exhibit more severe disease than those with a high prevalence of *S. epidermidis*, while disease progression can be linked to one dominant strain of *S. aureus* alone [247].

Early sequencing technologies greatly improved our understanding of microbial diversity but remained incapable of providing full microbial resolution or extrapolating phenotypic information. Development of next-generation sequencing has facilitated more detailed characterisation of the microbial landscape, thus revolutionising skin microbiome research [248]. Next-generation sequencing still utilises targeted amplification methods, such as 16S rRNA for bacteria, and internal transcribed spacer regions of the rRNA cistron for fungi [249], but it also enables metagenomic sequencing of entire genomes in a high-throughput, cost-effective manner [250]. Current metagenomic technologies include shotgun sequencing and newer “third generation” single-molecule long-read sequencing techniques (e.g., Oxford Nanopore and PacBio; comprehensively reviewed in [251]). Long-read sequencing (<10 kilobase pairs), although exhibiting higher error rates than short-read approaches, significantly improves taxonomic classification accuracy [252]. Nevertheless, individual platforms exhibit their own biases in taxa detection, making it difficult to discern true community structure and accurately compare findings of individual studies [253]. It has also been suggested that widely utilised sequencing techniques greatly overestimate species richness and diversity because they do not distinguish between extracellular DNA and DNA from viable bacteria [254]. To circumvent some of these issues, several studies are now adopting an integrated metagenomics approach to utilise multiple platforms, gaining the higher accuracy of short-read Illumina alongside the greater resolution of long-read Nanopore [255]. However, integrating workflows from entirely different platforms remains challenging [256], meaning this approach is not yet widely employed. Biases may even be resolved using mathematical modelling to identify the source of bias in workflows and correct them [253], whilst novel methods are being adopted to enable profiling of live bacterial DNA [254]. 

A major advantage of using metagenomics versus targeted amplification is that it can enable assembly of entire genomes, providing functional characterisation alongside higher taxonomic resolution, which is crucial given that different bacterial strains are associated with skin disease (e.g., *C. acnes* [257,258]). Metagenomic technologies have been utilised in assembling bacterial genomes to comprehensively profile the skin microbiome [259], along with identifying taxonomic and functional profiles of gut microorganisms in obesity [260]. Metagenomic sequencing also permits characterisation of the AMR gene reservoir of skin microbes (e.g., *S. epidermidis* [80]). Emerging single-cell approaches even have the potential to take this technology further, not only facilitating strain-level resolution, but also allowing investigation of the interactions between individual bacteria and colocalised bacteriophages [261]. These technologies will certainly be essential for the future development of targeted therapies to tackle AMR in skin and wound infections.

Despite the obvious advantages of metagenomics for microbiome research, full functional resolution and de novo assembly are frequently restricted to highly abundant strains due to the high read depth coverage required. This is particularly challenging in the skin, for which samples (e.g., swabs) often contain very small bacterial yields with varying microbial mass and high levels of host DNA contamination [262]. There is also a greater computational demand to assemble, map and analyse the data. To begin to address these challenges, statistical packages have been developed to utilise taxonomic and phylogenetic data to predict functional profiles and gene redundancies, and to identify biologically relevant differences between microbial profiles [263,264,265]. In addition, advanced techniques are improving read depth coverage (e.g., adaptive sequencing [266]) and increasing the reproducibility and translatability of microbiome research by standardising pipelines [267,268]. Indeed, combining these advances in microbial metagenomics with other technologies (e.g., RNA-Sequencing [269]) will enable a full system-level understanding of the role of the microbiome in skin health and disease.

### 8.2. Modelling the Skin Microbiome

It is becoming more apparent that the skin microbiome is crucially linked to cutaneous health and pathology [33,108]. Thus, suitable skin microbiome models are required for pre-clinical testing of potential therapies and cosmetics, alongside a better understanding of host–microbial interactions. However, development of suitable cutaneous microbiome models remains challenging due to the complexity and diversity of the microbiota and the dynamic interactions that exist between the host and microbial community [270,271]. Moreover, there is still a paucity of studies utilising skin models for microbiome research, with a large number instead focussing on pathogenic colonisation [272,273,274]. Key considerations in developing realistic microbiome models include providing a stable microbial community that represents normal skin microbiota and using models that faithfully mimic skin structure and physiology.

The simplest microbiome models provide a platform to elucidate fundamental relationships between individual bacteria and monolayers of host cells in vitro [275]. Although these models allow us to delineate specific host–bacterial interactions, such as how *S. aureus* can induce serine protease activity [276] and how *S. epidermidis* can activate epidermal defence in keratinocytes [277], they remain a reductionist approach that cannot recapitulate the complexities of a full microbial ecosystem in living skin. In addition, growing cells (bacterial and mammalian) in artificial culture alters their physiological responses, as cell culture media provides a very different nutritional source to natural skin milieu [278], and culture temperatures are several degrees higher than skin temperature [279]. To circumvent some of the challenges associated with in vitro culture systems, Van Der Krieken et al. developed a stratum corneum model to enable assessment of bacterial communities exposed to a more native substrate [280]. Interestingly, the authors demonstrated that bacterial diversity was maintained for up to seven days following inoculation. Nevertheless, this approach is still limited by the fact that the host component of the model is not viable. 

Three-dimensional human skin equivalents provide a more dynamic way to explore cutaneous physiology, comprising epidermal and dermal structures produced by skin cells. They more closely model natural cellular behaviours by providing a more native ECM and enabling true paracrine signalling between cells [281]. As stratification is an important component of skin physiology and host response, a major aim of developing 3D constructs is to create an epidermal barrier that closely resembles native skin, with differentiated layers and a similar composition of epidermal lipids [282]. These models can be useful to explore pathological host–microbe interactions, such as by demonstrating that reduced filaggrin expression increases *S. aureus* colonisation [283]. Human skin equivalents are also commercially available (reviewed in [284]), providing researchers with highly reproducible models requiring minimal additional resources. Skin equivalent models have even facilitated our understanding of the role of single and mixed communities of bacteria in regulating epidermal proliferation, differentiation and metabolism [271,285].

Despite their advantages, engineered skin constructs often lack more complex aspects of skin structure and physiology, such as glands, appendages, blood vessels and immune cells [202]. In recent years, three-dimensional skin constructs have been developed that harbour key components of innate skin physiology, such as vasculature [286,287], appendages [288] and improved immunocompetence [289,290]. This will not only advance the physiological relevance of these models, but simultaneously presents a more realistic environment with which to study the skin microbiome. For example, sebaceous glands provide a niche for *Cutibacterium*, a highly prevalent member of the native skin microbiota [60,85]. Thus, development of more sophisticated skin models incorporating sebaceous-like structures would greatly facilitate future microbiome studies.

By far, the most translational non-animal laboratory approach is to use living human skin, because it provides the advantage of preserving the intrinsic skin structure and cellular heterogeneity. Native skin maintains some immunological competence by virtue of containing resident immune cells that remain present for the first few days of culture [291,292]. Skin explants can be collected from a variety of body sites and donors spanning different ages, genders, ethnicities and pathological conditions to provide full representation of the human population [293]. Indeed, structural and functional changes in aged skin are associated with alterations in microbial composition [9,106], and ethnicity and gender can differentially alter skin pH [294], a key factor in determining microbial composition. Fresh human skin is an incredibly valuable resource for skin research but is often difficult to access and must be utilised within a specific window of viability. In addition, skin cultured ex vivo lacks a systemic response, which is particularly important when exploring the links between skin immunity and microbiota. A systemic influx of immune cells is also imperative for wound healing [295] and clearing pathogenic infection [296]. Indeed, novel culture methods, such as microfluidics, may circumvent some of these limitations by extending tissue viability ex vivo [297] and improving the immunocompetence of human skin equivalents [298].

Much of our understanding of the function of the microbiome has been derived from in vivo murine models that enable mechanistic insight into host–microbial interactions in complex living systems. In vivo skin microbiome research ranges from elucidating the role of commensal bacteria in skin development and maintenance [299,300] to evaluating how pathogenic dysbiosis leads to skin disease [10,29] and delays repair [37,226]. Murine models are also used to evaluate the efficacy of antimicrobial therapies against specific skin pathogens, such as *S. aureus* [301,302]. It is important to note, however, that there are fundamental differences in skin physiology, immunology and microbiology between mice and humans, which constrains the translatability of murine studies [303,304]. Moreover, there are a range of factors that can sway experimental outcomes, from animal strain and housing conditions to the type of infection model used [305,306,307], while the simultaneous underreporting of animal research protocols makes it difficult to extrapolate findings [308]. Pigs offer a more human-relevant model to assess skin physiology [309], but they are less tractable than rodents, and the porcine skin microbiome is less understood. Thus, future research should focus on developing methods to increase reproducibility and more faithfully represent the microbial diversity and complexity of human skin.

## 9. Future Directions

The rapid expansion of gut microbiome research has provided a conduit to explore the role of microbiota in other physiological systems, including the skin. An in-depth appreciation of the factors that shape the temporospatial distribution of cutaneous microbial communities could offer insight into the role of microbiota in skin ageing and pathology. Indeed, current research is often limited by correlative analysis and low taxonomic resolution; therefore, it is paramount that future investigations utilise cutting-edge sequencing methods and translatable models to provide real-world functional insight into host–microbe interactions in the skin.

Despite our limited understanding of the physiological role of skin microbiota in cutaneous biology, several strategies have been implemented to modulate the microbiome to improve health. This includes the use of antimicrobial products produced by bacteria to treat skin infection [148,310,311], and transplantation of commensal Gram-negative bacteria to ameliorate skin disease [49,312]. Moreover, the use of probiotics and postbiotics shows promise in alleviating ageing pathology [313,314,315] and accelerating wound repair [316,317] in experimental models. It would be fascinating to extend these findings to characterise how bacteria-derived treatments modulate the microbiome, alongside elucidating the potential microbial drivers of skin ageing and senescence-linked wound pathology. Wider adoption of sophisticated metagenomic technologies could facilitate functional characterisation of the microbiome and perhaps even provide a personalised approach to diagnosing and treating conditions underpinned by microbial dysbiosis.

In addition, AMR poses an urgent global healthcare challenge [318]; therefore, it is essential to develop non-antibiotic therapies to treat infection. It is now also appreciated that broad-spectrum antibiotics deplete the resident microbiota, contributing to the growth of AMR organisms [319,320]. Hence, emergent therapies must specifically target pathogens to ameliorate any impact on host commensals. One area that holds promise is the formulation of exogenously engineered bacteriophage-derived products capable of selectively killing specific skin pathogens (e.g., *S. aureus* [301]). However, we still require a greater understanding of the pathogenic landscape of skin, and the physiological mechanisms that promote pathogenicity, to utilise antimicrobials more effectively. Indeed, integrated system-level approaches may enable us to address this by providing a greater understanding of microbial community dynamics and delineating the functional relationships that exist between the skin and microbiota. Emerging computational methods could be utilised to integrate datasets and remove individual bias from different studies [253], therefore offering unprecedented insight into the global microbial landscape of the skin in health and disease. Computational methods may even be used to predict longitudinal progression in conditions underpinned by microbial dysbiosis [241], thus circumventing the need to collect longitudinal samples. Indeed, identification of the bacterial molecular signatures associated with certain disease states could even enable development of new and efficacious therapies for cutaneous pathologies and beyond.

## Figures and Tables

**Figure 3 ijms-24-03950-f003:**
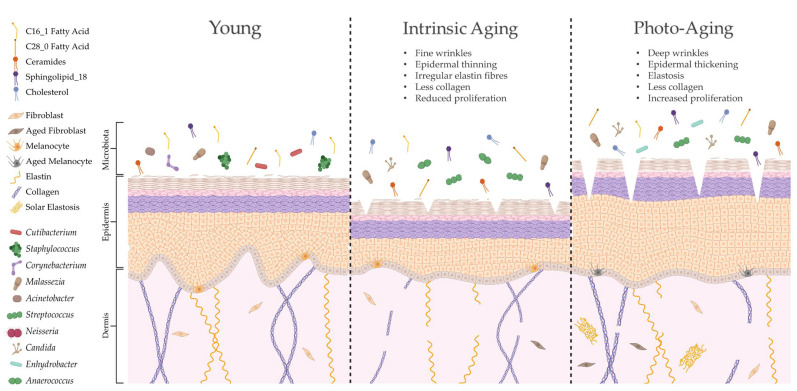
Ageing alters skin structure, function and microbial colonisation. Intrinsic ageing and photoageing cause differential alterations to cutaneous architecture and physiology, resulting in marked shifts in microbial composition [86,106]. Indeed, this altered skin microbiome may be shaped by specific modifications in lipid composition [9], which could further contribute to age-related cutaneous pathology. Created with biorender.com (accessed on 11 January 2023).

## Data Availability

Three search databases were used (PubMed, Google Scholar and Web of Science) to establish pertinent literature to remove biases associated with individual search engines. Key words were associated with relevant topics where applicable. We did not exclude publications based on date of publication, location of research or journal of publication. We based our assessment solely on research quality (as per the San Francisco Declaration on Research Assessment). We also did not search for specific methodologies (except for Section 8) as these can be indexed inconsistently.

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
