# Peer review of "The Skin Microbiome: Current Landscape and Future Opportunities"

_ijms, 2023, doi:10.3390/ijms24043950_

Round 1

Reviewer 1 Report

This review discusses the dynamic interactions between the skin and its resident microbes, highlighting key disparities and future revolutions to treat microbial dysbiosis. Overall, the review is well written, presenting good illustrations to explain the purpose of each section. I would like to congratulate the authors for writing this thoughtful review; thus, I recommend accepting the current version of the review. 

Author Response

We thank the reviewer for their comments and approval of our manuscript.

Reviewer 2 Report

This article deals with an exciting topic about skin microbiome and their advances in the field. The authors have an excellent review paper. However, it needs to be improved to be publishable in IJMS Journal.

My detailed comments and suggestions are in the attached file.

Author Response

Please see attached our responses to reviewers.

Round 2

Reviewer 2 Report

The authors addressed all my comments and suggestions correctly.

However, I would like to remark on my first comment to include the methodology to cover the literature used in their review.

This is useful for other readers because many review articles were published without a transparent procedure.

Author Response

We thank the reviewer for their comment. We have now added information around the review methodology in the Data Availability Statement.